# Endometriosis and Impaired Placentation: A Prospective Cohort Study Comparing Uterine Arteries Doppler Pulsatility Index in Pregnancies of Patients with and without Moderate-Severe Disease

**DOI:** 10.3390/diagnostics12051024

**Published:** 2022-04-19

**Authors:** Noemi Salmeri, Antonio Farina, Massimo Candiani, Carolina Dolci, Giulia Bonavina, Caterina Poziello, Paola Viganò, Paolo Ivo Cavoretto

**Affiliations:** 1Gynecology/Obstetrics Unit, IRCCS San Raffaele Scientific Institute, 20132 Milan, Italy; salmeri.noemi@hsr.it (N.S.); candiani.massimo@hsr.it (M.C.); dolci.carolina@hsr.it (C.D.); bonavina.giulia@hsr.it (G.B.); poziello.caterina@hsr.it (C.P.); 2Division of Obstetrics and Prenatal Medicine, Department of Medicine and Surgery (DIMEC), IRCCS Sant’Orsola-Malpighi Hospital, University of Bologna, 40138 Bologna, Italy; antonio.farina@unibo.it; 3Fondazione IRCCS Ca’ Granda Ospedale Maggiore Policlinico, 20122 Milan, Italy; paola.vigano@policlinico.mi.it

**Keywords:** endometriosis, US, placenta, pregnancy, obstetrics, maternal diseases, Doppler, uterine arteries, UtA-PI

## Abstract

The aim of this study was to evaluate if moderate-severe endometriosis impairs uterine arteries pulsatility index (UtA-PI) during pregnancy when compared to unaffected controls. In this prospective cohort study, pregnant women with stage III–IV endometriosis according to the revised American Fertility Society (r-AFS) classification were matched for body mass index and parity in a 1:2 ratio with unaffected controls. UtA-PIs were assessed at 11–14, 19–22 and 26–34 weeks of gestation following major reference guidelines. A General Linear Model (GLM) was implemented to evaluate the association between endometriosis and UtA-PI Z-scores. Significantly higher third trimester UtA-PI Z-scores were observed in patients with r-AFS stage III–IV endometriosis when compared to controls (*p* = 0.024). In the GLM, endometriosis (*p* = 0.026) and maternal age (*p* = 0.007) were associated with increased third trimester UtA-PI Z-scores, whereas conception by in-vitro fertilization with frozen-thawed embryo transfer significantly decreased UtA-PI measures (*p* = 0.011). According to these results, r-AFS stage III–IV endometriosis is associated with a clinically measurable impaired late placental perfusion. Closer follow-up may be recommended in pregnant patients affected by moderate-severe endometriosis in order to attempt prediction and prevention of adverse pregnancy and perinatal outcomes due to a defective late placental perfusion.

## 1. Introduction

Endometriosis is a chronic hormone-dependent, inflammatory, fibrotic condition in which endometrial stroma and epithelium can be identified outside the uterine cavity, predominantly but not exclusively in the pelvic cavity [1,2]. It is a common benign gynecological disease, affecting up to ~10% of women of reproductive age [3]. This condition causes dysmenorrhea, pelvic pain and infertility, often with a detrimental impact on quality of life and potentially leading to chronic comorbidities including malignancy [4,5]. An association between endometriosis and adverse obstetric and neonatal outcomes, such as a higher incidence of small for gestational age (SGA) fetuses, gestational hypertension and pre-eclampsia (PE) has recently emerged. This was suggested mainly as a consequence of defective deep placentation secondary to structural and functional abnormalities affecting the eutopic endometrium and myometrium of women with endometriosis [6,7,8,9,10]. Accordingly, endometriosis has been consistently found to be associated with an increased risk of placenta previa [6,7] and yet more recently, also an association between endometriosis and placenta previa placenta accreta spectrum disorders (PASD) has been demonstrated [11].

Major evidence from two decades of research showed that the combined screening test promoted by the Fetal Medicine Foundation (FMF) achieves the highest detection rate for some of these outcomes compared to any other available test [12]. As a consequence, its use has been recently endorsed by reference guidelines of leading international societies [13,14]. Such a test is based upon a combination of maternal demographic characteristics and biomarkers, of which uterine arteries Doppler (UtA) for pulsatility index (PI) measurement is undoubtedly the milestone [15,16]. Remarkably, UtA-PI was shown to be an effective and reliable tool for early detection of PE, fetal growth restriction (FGR) and SGA fetuses and was, therefore, incorporated into obstetric care [13,14,17,18]. 

Previous researchers failed to show significant differences in measurements of UtA-PI, pregnancy associated plasma protein-A (PAPPA-A) as well as rates of SGA and PE between pregnant women with and without adenomyosis, except for cases with diffuse disease [19]. However, the latter study failed to assess gestational age (GA) corrected standardized values of UtA-PI, and did not consider significant confounders potentially affecting UtA-PI, such as in-vitro fertilization (IVF) conception [20,21].

The aim of the current study was to evaluate whether the presence of stage III–IV endometriosis according to the revised American Fertility Society classification (r-AFS) [22] could impair UtA-PI measures when compared to controls with no evidence of the disease. This may contribute to elucidating the role of endometriosis in increasing the risk of placental dysfunction based on a measurable late defective uterine perfusion.

## 2. Materials and Methods

### 2.1. Study Design and Setting

This is an observational, monocentric, prospective cohort study carried out from January 2016 to January 2021 at IRCSS San Raffaele Scientific Institute in Milan, Italy. The study was conducted and reported according to the Strengthening the reporting of observational studies in epidemiology (STROBE) guidelines [23]. 

### 2.2. Population

Singleton pregnancies with moderate or severe endometriosis (stage III or IV following r-AFS classification) assessed either surgically and histologically or confirmed by ultrasound (US) were selected as cases. Endometriotic lesions were phenotypically classified according to their localization, as ovarian endometrioma (OMA), deep endometriosis (DE), or both [24,25]. No cases with superficial peritoneal endometriosis (SPE) or with adenomyosis were enrolled. Surgical and histological reports were revised at the time of cases enrolling. US diagnoses of DE or OMA with no surgical treatment were performed uniquely in our Institute by expert US operators in this field. 

Pregnant women with no evidence of the disease (no previous diagnosis of endometriosis, availability of a normal transvaginal US before pregnancy and a negative history of dysmenorrhea, deep dyspareunia and/or chronic severe pelvic pain) were randomly enrolled as controls in a 1:2 match. Matching of cases to controls was achieved by enrolling the first two pregnancies with no evidence of endometriosis, with identical parity and body mass index (BMI) similarity (±1 unit), observed within 5 days from recruitment of each case with endometriosis. The patient selection process is shown in Figure 1. 

Women with multiple gestations, fetal structural abnormality or fetal aneuploidy, uterine malformation, significant uterine fibroids, previous uterine surgery and significant pre-gestational maternal diseases, such as cardiovascular, liver or renal disease, diabetes mellitus, coagulation and autoimmune disorders were excluded. 

### 2.3. Data Collection

Data collection followed the principles outlined in the Declaration of Helsinki [26]. Pregnancies were dated according to the last menstrual period (LMP) in case of spontaneous conception and according to a pseudo-LMP fixed nineteen days before the blastocyst transfer or seventeen days before cleavage-stage embryo transfer in the case of IVF conception. In all cases, GA was confirmed by measuring the crown–rump length (CRL) in the first trimester by US assessment following the reference guidelines [27,28]. UtA-PI was assessed by Doppler US at 11–14, 19–22 and 26–34 weeks of gestation following the Fetal Medicine Foundation (FMF) and the International Society of Ultrasound in Obstetrics and Gynaecology (ISUOG) guidelines [13,29,30]. UtA-PI was calculated as (peak systolic velocity–end diastolic velocity)/time averaged velocity; left and right UtA-PI measurements were averaged to estimate a mean and compared to published reference ranges [31]. All the Doppler US assessments were performed by operators with extensive experience and expertise holding the certification of competence in Doppler US granted by the FMF, using top level machines (Samsung WS80) equipped with multi-frequency convex transabdominal transducers (Samsung, Seoul, Korea).

Patients’ data were prospectively recorded in a database including the following: (i) baseline maternal characteristics, such as age, BMI, parity, method of conception, smoking status; (ii) first, second and third trimester US measurements including CRL, serum levels of free-beta human chorionic gonadotropin (beta-hCG) and PAPP-A, estimated fetal weight (EFW); (iii) maternal and perinatal outcomes, including GA at delivery (weeks, days), birthweight (grams, centiles, Z-scores), the proportion of SGA and large for gestational age (LGA) fetuses, preterm birth (PTB), gestational hypertension/PE and oligohydramnios. Estimated fetal weight (EFW) centiles were defined according to FMF fetal and neonatal population weight charts [31]. According to Delphi consensus criteria [32], SGA was defined as a birthweight below the 10th centile for gestational age. LGA indicates a birthweight greater than the 90th centile for GA according to large population-based standards. As for World Health Organization (WHO) recommended definitions, PTB was defined in the setting of a delivery that occurred before the completion of 37 gestation weeks [33].

### 2.4. Outcome

The outcome of this study was UtA-PI Z-scores in the 1st, 2nd and 3rd trimester of pregnancy calculated from reference equations of previously published normal ranges [34].

### 2.5. Statistical Analysis

The Shapiro–Wilk test was used to ascertain whether continuous variables had a normal distribution. Continuous and normally distributed variables were presented as mean ± standard deviation (SD), whereas continuous not normally distributed variables were expressed as the median and interquartile range (IQR) and categorical variables were presented as absolute values and percentages (%). The Student *t*-test or Wilcoxon signed-ranks test with an exact test for quantitative variables and Pearson’s Chi square test or Fisher’s exact test for qualitative variables were performed to compare data of cases and controls, as appropriate.

A General Linear Model (GLM) was performed to determine the association between endometriosis and UtA-PI Z-scores. Analysis of minimal deviance was used to define optimal link and variance functions. Conventional goodness-of-fit tests were performed to select the best multivariable model. Average marginal effects with 95% Confidence Intervals (CIs) for all the variables included in the model were additionally plotted. STATA version 17 software (Stata Corp LLC, 2021, College Station, Brazos County, TX, USA) was used for statical analysis. All tests were two-sided and *p*-values < 0.05 were considered statistically significant.

### 2.6. Power Analysis

Power analysis for the comparison of two independent means was conducted before the enrollment started, setting the sample allocation ratio to 1:2. The sample size calculation analysis indicated that a total of 32 cases and 64 controls, with a UtA-PI coefficient of variation equal to SD/mean ≤0.35 as calculated by published data [34], were required to detect up to 10% change in the outcome measure (UtA-PI Z-score), with a power of 0.8 and a type I error of 0.05.

## 3. Results

### 3.1. Study Population

Of the 47 endometriosis patients enrolled in the study, 36 (76.6%) had an OMA phenotype, 1 (2.1%) had a DE phenotype and 10 (21.3%) had simultaneously an OMA and a DE localization. In four out of 46 cases with an ovarian localization of the disease (8.7%), bilateral endometriomas were reported. Additionally, 27/47 (57.4%) of endometriosis cases underwent a previous laparoscopic and histologic confirmation of the disease, whereas, in 20/47 (42.6%) of cases, DE or OMA were assessed before conception by US. All the included cases had a stage III–IV (i.e., moderate-severe) endometriosis disease according to r-AFS classification. The study group consisted of 47 (33.6%) cases of endometriosis and 93 (66.4%) controls with no evidence of the disease, matched by parity and BMI. 

### 3.2. Baseline Characteristics and Univariable Analysis

Baseline characteristics of endometriosis cases and controls are shown in Table 1. As per controls-matching, no significant differences in BMI and parity (nulliparous vs. parous) were observed. Additionally, maternal age, cigarette smoking status and type of conception (spontaneous vs. IVF) did not differ between the two groups. IVF was from cycles with frozen-thawed embryo transfer in 87.5% (95% CI: 74.4%–94.4%) of the cases. 

First, second and third trimester US variables assessment in cases and controls are summarized in Table 2. There were no statistically significant differences both in the 1st or in the 2nd trimester UtA-PI Z-scores between the two groups; conversely, 3rd trimester UtA-PI Z-scores were significantly higher in endometriosis (median: 0.03; IQR: −0.02 to 0.22) than in controls (median: −0.04; IQR: −0.09 to 0.1) (*p* = 0.024). No statistically significant differences were observed either in the 1st trimester levels of serum biomarkers (MoM PAPP-A and MoM free β-hCG) or in CRL standardized measures. A higher yet not significant rate of SGA fetuses and SGA neonates were observed in cases; on the other hand, a lower still not significant proportion of LGA fetuses and LGA neonates was observed in cases. Remarkably, no significant differences in GA at US examinations (during 1st, 2nd, 3rd trimesters) were observed. 

Pregnancy and perinatal outcomes in cases and controls are presented in Table 3. No differences were observed comparing pregnancy and perinatal outcomes of cases and controls with the exception of a higher proportion of PTB in the endometriosis group (6/47; 12.8%) compared to controls without endometriosis (3/93; 3.2%) (*p* = 0.03), despite no difference in overall GA at delivery between the two groups. 

### 3.3. General Linear Model

To further investigate the observed significant differences of third trimester UtA-PI Z-scores in endometriosis and controls, we performed a GLM (Table 4). 

In the univariate model, the presence of endometriosis was associated with a significant increase by 0.111 units (95% CI: 0.016–0.206) in third trimester UtA-PI Z-scores (*p* = 0.021). 

The association between third trimester UtA-PI Z-scores (response variable) and endometriosis was further assessed with a multivariable model adjusted for several explanatory variables including nulliparity, conception by IVF, maternal age and BMI. Average marginal effects with 95% CIs for all the independent variables included in the model are shown in Appendix A. The multivariable GLM showed an increase in third trimester UtA-PI Z-scores in endometriosis (*β* = 0.0132; 95% CI: 0.016 to 0.248; *p* = 0.026). Similarly, a significant increase in the response variable was observed with increasing maternal age (*p* = 0.007). Conversely, conception by IVF significantly decreased third trimester UtA-PI Z-scores (*p* = 0.011). Neither nulliparity nor BMI significantly affected third trimester UtA-PI Z-scores. A 35.797 log likelihood was computed for the proposed model. Post-estimation goodness of fit analysis for the multivariate model showed an Akaike’s information criterion (AIC) of −59.595 and a Bayesian information criterion (BIC) based on the number of observations was equal to −47.232. 

Box plots of third trimester UtA-PI Z-scores in endometriosis patients and controls weighted for the predictors included in the multivariable model are shown in Figure 2. 

## 4. Discussion

This prospective cohort study showed significantly higher third trimester UtA-PI Z-scores in patients with r-AFS stage III–IV endometriosis when compared to unaffected controls and no differences in first or in second trimesters UtA-PI Z-scores. The extent of UtA-PI increase in endometriosis was about 13% with additional effects of maternal aging (positive by 1.5%) and IVF conception (negative by 11%). 

We would like to speculate on the underlying mechanisms on the basis of our findings. In normal pregnancies, UtA-PI progressively declines with advancing gestation, reflecting the underlying process of placentation and so the conversion of spiral arteries into uteroplacental arteries [35]. Such a decline of UtA-PIs emphasizes the role of maternal hemodynamic changes taking place up to the third trimester. 

Endometriosis-related fibrosis may interfere with the regulation of peripheral vessels physiology altering endothelial function and vascular stiffness [36,37]. The fibrotic entrapment of pelvic vessels may determine an increase in the arteries’ wall stiffness and a consequent decrease in vascular compliance, thus hampering physiological hemodynamic adaptation occurring in late pregnancy. In this setting, higher uterine arteries’ resistance in endometriosis rather than in controls may become clinically measurable only during the third trimester of pregnancy due to the concomitant effects of maternal hemodynamic remodeling.

The physiological progressive decrease of UtA-PIs throughout gestation is thought to be also due to a progressive increase in estrogen levels during pregnancy, which tend to have a vasodilatory effect [38]. We can suppose that the hyper-estrogenic milieu known to characterize the pelvis of patients with endometriosis [39] could lead to dysregulation of uterine vessels’ response to estrogens during pregnancy. Indeed, the actions of estrogen and progesterone during pregnancy have a sequential pattern, so those hormones are tightly and reciprocally controlled through the regulated expression of steroid receptors, chaperone proteins and downstream signaling components. Abnormal estrogen and progesterone receptors mediated signaling pathways in endometriosis have been suggested to drive the endometrial dysfunction responsible for aberrant chorion–decidua interactions during late pregnancy [40].

Remarkably, a recent review emphasized the possibility of defective deep placentation in endometriosis due to several structural and functional abnormalities of eutopic endometrium and myometrium [41]. As a matter of fact, the presence of ectopic tissue in endometriosis is associated with local overproduction of several pro-inflammatory and pro-fibrotic cytokines and chemokines (i.e., interleukin 1beta (IL-1β), IL-6, tumor necrosis factor-alpha (TNF-α), transforming growth factor-beta (TGF-β), monocyte chemotactic protein 2 (MCP-2)) [42,43,44,45] and yet also aberrant levels of coagulation and inflammatory parameters in the peripheral blood of endometriosis have been observed [46]. Increased local and systemic inflammatory pathways in endometriosis are considered a major causal factor for explaining the immune and vascular dysfunction in placenta/decidua interactions, leading to PE, FGR and PTB. Indeed, the inflammatory environment induced by endometriosis alters endometrial progesterone response and myometrial decidualization, determining an abnormal trophoblastic invasion into the myometrial junctional zone during pregnancy [6]. In physiological pregnancies, the spiral arteries replace their musculo-elastic wall with amorph fibrinoid coating containing trophoblast cells, leading to a high-flow, low-resistance, non-vasoactive vessel capable of supplying sufficient blood to meet the increasing demands of the developing fetus [47]. In pregnancies complicated by PE and FGR but possibly even in those complicated by endometriosis, spiral arteries maintain their musculo-elastic layer and often undergo hyperplastic changes, with a consequent increase in uterine arteries’ impedance to flow and placentation defects [8]. 

The results of the present study suggest that, in line with the higher risk of adverse maternal/perinatal outcomes observed in large population-based studies, the higher uterine artery impedance in the third trimester of pregnancies with documented r-AFS stage III–IV endometriosis may be a clinically measurable indicator of impaired placentation in those patients [48]. The finding of normality in first trimester serum biochemistry in our patients (free beta-hCG and PAPP-A) somehow confirms the finding of normal UtA-PI in the first trimester. Hence, the clinical manifestation of placental dysfunction may be a slow process occurring with a gradual progression from an early subclinical phase to the late phase in which UtA-PI changes become clinically measurable and the final pregnancy outcome becomes obvious.

Endometriosis was found to be associated with several adverse obstetric/perinatal outcomes [49]. Two recent meta-analyses comprising 1,924,114 and 2,517,516 women, respectively, showed that endometriosis patients have a higher risk of miscarriage, PTB, placenta previa and SGA infants [50,51]. Furthermore, other two systematic reviews remarked on the association between endometriosis and other maternal/perinatal complications, such as gestational hypertension or PE, gestational diabetes, antepartum hemorrhage, antepartum hospital admissions, labor dystocia and cesarean section, preterm premature rupture of membranes, neonatal intensive care unit admission even up to stillbirth and neonatal death [52,53]. This available meta-analytic evidence, however, as stated by their authors, is limited by the statistical and clinical heterogeneity, especially in diagnostic criteria, classification and reporting systems of endometriosis disease [24]. Our findings confirmed the previously reported association between endometriosis and increased risk of PTB, which seems to be best justified by the well-known hyperactivation of inflammatory and immune systems in endometriosis [54]. 

According to previous evidence, our data also support the additional effect of increasing maternal age on UtA-PIs in the third trimester of pregnancy [55]. Advanced maternal age has already been associated with multiple adverse pregnancy outcomes [56] and seems to have multifactorial pathogenesis, mostly accounted for by general reduced vascular compliance and cardiovascular adaptation in older women [57,58]. Furthermore, conception by IVF was found to be associated with significantly decreased third trimester UtA-PIs. Since the vast majority of cases from our current series are from frozen-thawed embryo transfers, this is in agreement with previous major evidence from our group [20,21], recently confirmed by other authors [59].

This study presents some limitations. Firstly, the diagnosis of deep and ovarian endometriosis was in some cases assessed by the US alone before conception, even if in the majority of cases a diagnostic laparoscopy was performed. However, US assessment for deep and/or ovarian endometriosis has been validated worldwide as a reliable diagnostic tool [60], especially when performed by expert operators, such as in our Institute. Secondly, all enrolled cases had r-AFS stage III–IV disease, thus suggesting that our results could be a reliable estimate only for advanced stages of endometriosis and may not be generalizable to minimal or moderate endometriosis (r-AFS stage I–II). Thirdly, the unlikely presence of superficial peritoneal endometriosis in controls cannot be totally excluded. However, given the selection criteria of controls (negative US and no history of dysmenorrhea, deep dyspareunia and/or chronic severe pelvic pain), the presence of unknown endometriosis is very unlikely. Yet even if previous studies [19] have demonstrated significant differences in mean UtA-PIs between patients with diffuse adenomyosis and controls, we did not include adenomyosis in our work because of the well-known pathophysiological difference between endometriosis itself and adenomyosis [61]. 

However, this study presents several strengths, as well. Firstly, the case-control matching for parity and BMI (known confounders altering UtA-PI), with no differences in the type of conception between the two groups, allowed us to control for potential sources of biases. Secondly, data were collected with quality and homogeneity of ultrasound methodology (all US measurement were performed in a single-centre by certified operators with homogeneous protocols and reduced inter-operators variability). Thirdly, the robustness of the statistical analysis performed within a model providing a high power contributed to strengthening our conclusions. Finally, restricting the analysis to the moderate and severe forms of endometriosis (r-AFS stage III–IV) contributed on the other hand to reduce the clinical heterogeneity of the study group.

## 5. Conclusions

The presence of stage III–IV endometriosis according to r-AFS is associated with a clinically measurable impaired placental perfusion during the third trimester of pregnancy, as showed by the 13% higher UtA-PI Z-scores as compared to unaffected controls. Endometriosis-related fibrosis altered vascular response to estrogens/progesterone and the inflammatory microenvironment may be the pathophysiological mechanisms justifying our findings. Thus, we feel appropriate to recommend closer follow-up with serial well-being and growth scans in the third trimester (28 to 36 weeks) of patients with advanced stages of endometriosis, in order to diagnose or prevent potential consequences of defective placentation. Further studies will be needed to elucidate the potential link between impaired placental perfusion and endometriosis in relation to obstetric/perinatal complications, with the final aim to improve the final pregnancy outcome. 

## Figures and Tables

**Figure 1 diagnostics-12-01024-f001:**
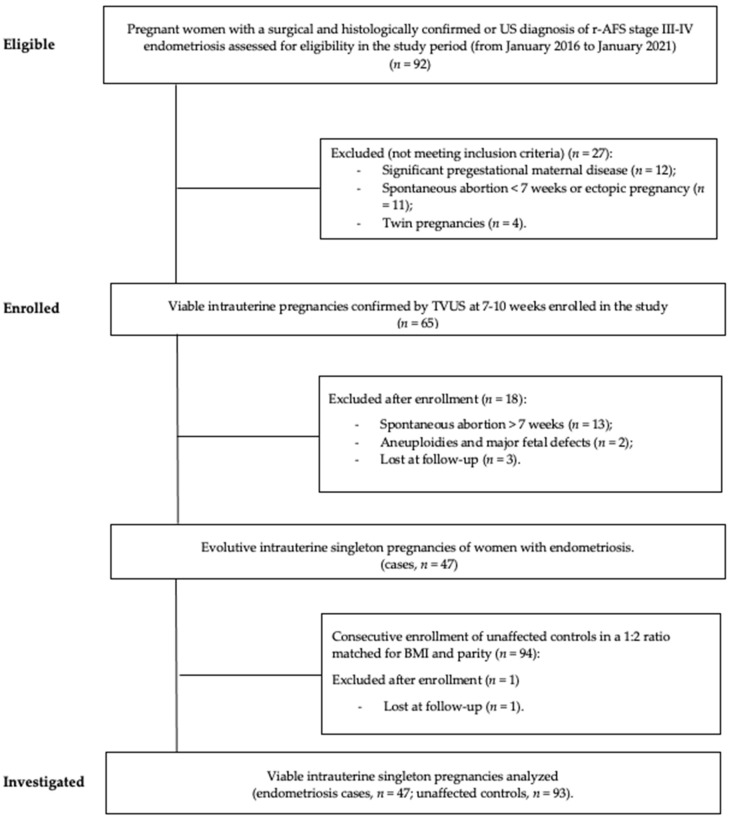
Strengthening the reporting of observational studies in epidemiology (STROBE) flow chart of study design. Abbreviations: US, ultrasound; TVUS, transvaginal ultrasound; BMI, body mass index.

**Figure 2 diagnostics-12-01024-f002:**
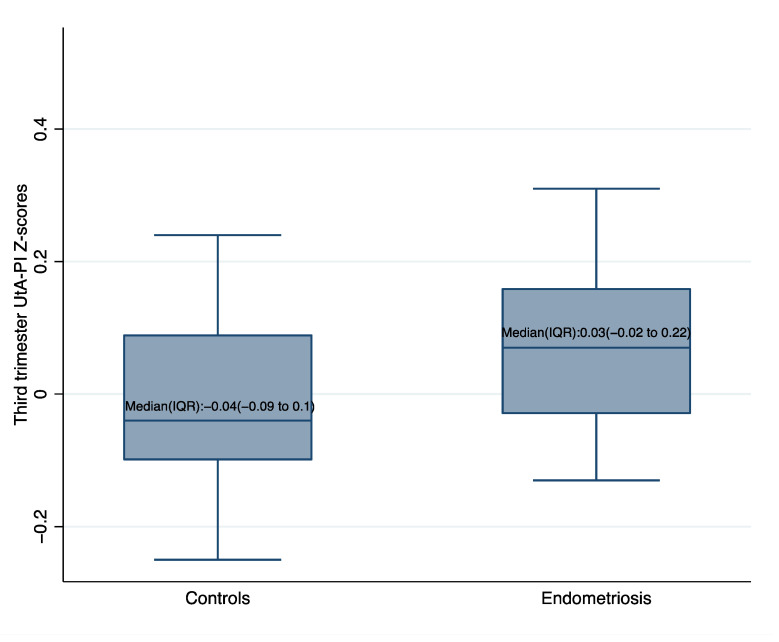
Box plots of third trimester UtA-PI Z-scores in endometriosis and controls. weighted for the predictors included in the adjusted general linear model (GLM). Notes: weighting predictors included in the model: conception by in vitro fertilization (*p* = 0.011) and maternal age (*p* = 0.007). Abbreviations: UtA-PI, uterine artery pulsatility index; IQR, interquartile range.

**Table 1 diagnostics-12-01024-t001:** Baseline characteristics of cases and controls (*n*= 140).

	Cases (*n* = 47)	Controls (*n* = 93)	*p*-Value
Maternal age, years ^1^	34.02 ± 4.9	35.22 ± 4.2	0.139
BMI, kg/m^2^ ^2^	20.7 (19.4 to 22)	21.04 (19.1 to 22.7)	0.672
Smoking, *n* ^3^	3 (8.8%)	6 (6.5%)	0.656
Nulliparous, *n* ^3^	33 (70.2%)	65 (69.9%)	0.969
Spontaneous conception, *n* ^3^	30 (63.8%)	62 (66.7%)	0.738
IVF, *n* ^3^	17 (36.2%)	31 (33.3%)	0.738

Notes: Data are ^1^ mean ± SD, ^2^ median (IQR) or ^3^
*n* (%). Abbreviations: BMI, body mass index; IVF, In Vitro Fertilization.

**Table 2 diagnostics-12-01024-t002:** First, second and third trimester US variables assessment in cases and controls.

	Cases (*n* = 47)	Controls (*n* = 93)	*p*-Value
First trimester			
GA, weeks + days ^2^	11 + 3 (11 + 2 to 12 + 4)	11 + 3 (11 + 2 to 12 + 3)	0.366
CRL, mm ^1^	58.19 ± 8.19	59.26 ± 7.36	0.441
CRL Z-score ^2^	0.51 (0.01 to 0.84)	0.50 (−0.28 to 1.17)	0.952
UtA PI ^2^	1.63 (1.02 to 1.79)	1.51 (1.14 to 1.86)	0.715
UtA PI Z-score ^2^	−0.04 (−0.51 to 0.08)	−0.11 (−0.39 to 0.17)	0.462
MoM free β-hCG ^2^	0.99 (0.81 to 1.75)	0.94 (0.63 to 1.32)	0.326
MoM PAPP-A ^2^	1.01 (0.83 to 1.48)	1.13 (0.82 to 1.51)	0.837
Second trimester			
GA, weeks + days ^2^	20 + 3 (19 + 2 to 21 + 4)	20 + 3 (20 + 2 to 21 + 3)	0.681
UtA PI ^2^	0.97 (0.75 to 1.12)	0.85 (0.74 to 1.14)	0.741
UtA PI Z-score ^2^	−0.05 (−0.25 to 0.4)	−0.18 (−0.27 to 0.06)	0.957
Third trimester			
GA, weeks + days ^2^	31 + 3 (30 + 2 to 32 + 4)	30 + 3 (30 + 3 to 31 + 3)	0.312
UtA PI ^2^	0.87 (0.74 to 0.90)	0.73 (0.63 to 0.88)	**0.033**
UtA PI Z-score ^2^	0.03 (−0.02 to 0.22)	−0.04 (−0.09 to 0.1)	**0.024**
EFW, grams ^2^	1900 (1582 to 2123)	1665 (1297 to 2086)	0.325
EFW, centiles ^2^	54.72 (28.25 to 83.75)	57.18 (32.75 to 84)	0.679
SGA fetuses ^3^	4 (8.5%)	3 (3.2%)	0.224
LGA fetuses ^3^	4 (8.5%)	13 (14%)	0.422

Notes: Data are ^1^ mean ± SD, ^2^ median (IQR) or ^3^
*n* (%). Abbreviations: US, ultrasound; GA, gestational age; CRL, crown-rump length; UtA PI, uterine artery pulsatility index; MoM, multiples of the normal median; free β-hCG, free β-human chorionic gonadotrophin; PAPP-A, Pregnancy-associated plasma protein A; EFW, estimated fetal weight; SGA, small for gestational age; LGA, large for gestational age.

**Table 3 diagnostics-12-01024-t003:** Baseline characteristics of cases and controls (*n* = 140).

	Cases (*n* = 47)	Controls (*n* = 93)	*p*-Value
Pregnancy			
Gestational hypertension ^2^	3 (6.4%)	2 (2.2%)	0.334
Pre-eclampsia ^2^	1 (2.12%)	0 (0%)	0.336
Placenta previa ^2^	2 (4.3%)	0 (0%)	0.111
Oligohydramnios ^2^	0 (0%)	5 (5.4%)	0.168
Perinatal			
GA at delivery, weeks + days ^1^	38 + 2 (37 + 2 to 39 + 3)	38 + 2 (38 + 2 to 39 + 3)	0.998
Preterm birth (<37 weeks) ^2^	6 (12.8%)	3 (3.2%)	**0.030**
Birth weight, grams ^1^	3050 (2870 to 3370)	3225 (2975 to 3450)	0.304
Birth weight, centiles ^1^	34 (13 to 52)	42 (25 to 65.5)	0.223
Birth weight, z-score ^1^	−0.41 (−1.15 to 0.06)	−0.19 (−0.67 to 0.40)	0.192
SGA neonates ^2^	7 (14.9%)	9 (9.7%)	0.404
LGA neonates ^2^	1 (2.1%)	5 (5.4%)	0.664
5 min Apgar score < 7 ^2^	3 (6.4%)	2 (2.2%)	0.334
10 min Apgar score < 7 ^2^	1 (2.1%)	0 (0%)	0.336
Fetal sex, male/female ^2^	47/42 (52.8%)	22/17 (56.4%)	0.707

Notes: Data are ^1^ median (IQR), ^2^
*n* (%). Abbreviations: SGA, small for gestational age; LGA, large for gestational age.

**Table 4 diagnostics-12-01024-t004:** General Linear Model (GLM) exploring the association between third trimester UtA-PI Z-scores (response variable) and endometriosis.

Parameter	Beta Coefficient	SE	95% CI	*p*-Value
Endometriosis	0.132	0.059	0.016 to 0.248	**0.026**
Conception by IVF	−0.109	0.043	−0.193 to −0.025	**0.011**
Maternal age, years	0.013	0.005	0.003 to 0.022	**0.007**
Nulliparity	0.061	0.044	−0.024 to 0.146	0.160
BMI, kg/m^2^	−0.008	0.007	−0.021 to 0.006	0.264

Abbreviations: UtA-PI, uterine artery pulsatility index; SE, Standard Error; 95% CI, 95% Confidence Intervals; IVF, In Vitro Fertilization; BMI, body mass index.

## Data Availability

The data supporting the findings of this study are available from the corresponding author upon reasonable request. The preliminary results of this study were object of oral presentations in the last congresses of the International Society of Ultrasound in Obstetrics and Gynecology (ISUOG) and Italian Society of Ultrasound in Obstetrics and Gynecology (SIEOG), in the year 2021.

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
