# Peer review of "Endometriosis and Impaired Placentation: A Prospective Cohort Study Comparing Uterine Arteries Doppler Pulsatility Index in Pregnancies of Patients with and without Moderate-Severe Disease"

_diagnostics, 2022, doi:10.3390/diagnostics12051024_

Round 1
Reviewer 1 Report
In this study the authors evaluated if moderate-severe endometriosis could impair UtA-PI during pregnancy compared to unaffected controls. UtA-PIs were assessed at 11–14, 19–22 and 26–34 weeks of gestation. They found an higher third trimester UtA-PI Z-score in patients with r-AFS stage III-IV endometriosis compared to controls. In the GLM, endometriosis and maternal age were associated with increased third trimester UtA-PI Z-scores, whereas conception by in-vitro fertilization with frozen-thawed embryo transfer significantly decreased UtA-PI measures concluding that r-AFS stage III-IV endometriosis was associated with a clinically measurable impaired late placental perfusion.
- The aim of this study was to evaluate whether the presence of stage III-IV endometriosis could impair UtA-PI measures when compared to controls with no evidence of the disease. This is an original and relevant topic that may contribute to elucidate the role of endometriosis in increasing the risk of placental dysfunction using UtAPI, a useful tool to measure uterine perfusion.
- Although UtAPI is a diagnostic tool normally used in gynaecology, the association with endometriosis as a risk of pregnancy complications is very interesting.
- The study is well designed and the cohort analysed is statistically valid (93 controls and 47 cases)
- The conclusions made by the authors are adequate to the results found in the cohort.
- Although generally discussed in lines 295-300, a deeper discussion on the complexity of inflammatory environment found in endometriosis would also explain alterations in UtAPI.
For example, it has been reported that endometriosis is characterised by increased levels of TGFB (PMID: 26708185) and TNF-α (PMID: 33152545) that can impair placentation. However, neither these or other cytokines are mentioned in the manuscript.
The manuscript is clear, generally well written and can be accepted in the present form.
Author Response
REVIEWER 1:
Comment of Reviewer 1:
In this study the authors evaluated if moderate-severe endometriosis could impair UtA-PI during pregnancy compared to unaffected controls. UtA-PIs were assessed at 11–14, 19–22 and 26–34 weeks of gestation. They found an higher third trimester UtA-PI Z-score in patients with r-AFS stage III-IV endometriosis compared to controls. In the GLM, endometriosis and maternal age were associated with increased third trimester UtA-PI Z-scores, whereas conception by in-vitro fertilization with frozen-thawed embryo transfer significantly decreased UtA-PI measures concluding that r-AFS stage III-IV endometriosis was associated with a clinically measurable impaired late placental perfusion. The aim of this study was to evaluate whether the presence of stage III-IV endometriosis could impair UtA-PI measures when compared to controls with no evidence of the disease. This is an original and relevant topic that may contribute to elucidate the role of endometriosis in increasing the risk of placental dysfunction using UtAPI, a useful tool to measure uterine perfusion. Although UtAPI is a diagnostic tool normally used in gynaecology, the association with endometriosis as a risk of pregnancy complications is very interesting. The study is well designed and the cohort analysed is statistically valid (93 controls and 47 cases). The conclusions made by the authors are adequate to the results found in the cohort. Although generally discussed in lines 295-300, a deeper discussion on the complexity of inflammatory environment found in endometriosis would also explain alterations in UtAPI. For example, it has been reported that endometriosis is characterised by increased levels of TGFB (PMID: 26708185) and TNF-α (PMID: 33152545) that can impair placentation. However, neither these or other cytokines are mentioned in the manuscript. The manuscript is clear, generally well written and can be accepted in the present form.
Reply to Reviewer 1:
We warmly thank Reviewer 1 to have appreciated our manuscript and our effort in presenting this novel topic. According with Reviewer 1, this is the first study investigating the role of endometriosis in increasing the risk of placental dysfunction using UtAPI measurement. In light of its novelty, to fully justify our findings, we have discussed comprehensively all the several pathogenetic pathways potentially involved in this complex process. We do believe that endometriosis-related fibrosis may be the leading mechanical process of increased uterine arteries’ impedance to flow, yet we have also speculated on the theoretically explanatory functional mechanisms lying underneath placentation defects in endometriosis. As a matter of fact, in our Discussion section we have argued the role of estrogen and progesterone imbalance yet also of local and systemic inflammatory pathways in endometriosis to justify the immune and vascular dysfunction in placenta/decidua interactions during pregnancy. According with Reviewer 1, endometriosis is a pelvic chronic inflammatory condition and yet pro-inflammatory and pro-fibrotic cytokines and chemokines may provide new inroads into the deep comprehension of the defective placentation observed in endometriosis. We have remarked that those mechanisms have been previously suggested by other researchers to explain the increased risk of pre-eclampsia, fetal growth restriction and pre-term birth observed in endometriosis. Following Reviewer 1 suggestions, we have also modified the original manuscript mentioning some of the inflammatory pathways and aberrant cytokines expression observed in endometriosis and potentially linked with placentation defects [Lines 293-298]. References of suggested published articles have been added accordingly.
Reviewer 2 Report
The authors examined the effect of endometriosis on the uterine artery pulsatility index (UtA-PI) during pregnancy. This manuscript is well written; however, I have several suggestions to improve the manuscript. The reviewer’s comments are as follows.
Line 43
A recent systematic review examined the relationship between endometriosis and the placenta accreta spectrum of disorder (PASD). The authors may add the PASD as an adverse obstetric outcome related to endometriosis.
It is unclear to me why the authors think endometriosis affects the value of UtA-PI. Please indicate your hypothesis to provide the rationale for this study.
Line 75
Please clarify how the authors diagnose severe endometriosis by using ultrasonography. Please show the number of histologically confirmed endometriosis.
Did the authors examine the effect of non-severe endometriosis on the value of UtA-PI?
Additional comments:
I think the topic of this study is relevant and interesting. However, the authors need to clarify why the authors conducted this study.
Compared with other published material this topic has not been determined and the results of this study are unique.
The methodology appeared to be appropriate.
Yes, the conclusion is consistent with the evidence and argument presented. The authors discussed the main question well.
Reference wise, the authors need to add some recent studies (systematic review) regarding the endometriosis.
The tables and figures appeared to be appropriate and easy to understand.
Author Response
REVIEWER 2:
Comments of Reviewer 2:
The authors examined the effect of endometriosis on the uterine artery pulsatility index (UtA-PI) during pregnancy. This manuscript is well written; however, I have several suggestions to improve the manuscript. The reviewer’s comments are as follows.
Question:
Line 43
A recent systematic review examined the relationship between endometriosis and the placenta accreta spectrum of disorder (PASD). The authors may add the PASD as an adverse obstetric outcome related to endometriosis.
Reply: We warmly thank Reviewer 2 for the suggestion. As a matter of fact, despite endometriosis has been found to be associated with an increased risk of placenta previa by many researches, the systematic review cited by the Reviewer 2 is the first suggesting an association between endometriosis and placenta previa placenta accreta spectrum disorders (PASD). According with the Reviewer suggestion, we have added such a novel discovery to remark the defective placentation observed in endometriosis [Lines 47-50].
Question:
It is unclear to me why the authors think endometriosis affects the value of UtA-PI. Please indicate your hypothesis to provide the rationale for this study.
Reply: As mentioned in our Introduction section [Lines 40-51] and further discussed in our Discussion section [Lines 364-377], endometriosis, especially in the presence of deep and ovarian lesions, has been found to be consistently associated with several adverse maternal and obstetric outcomes. Since uterine arteries Doppler (UtA) for pulsatility index (PI) measurement has been validated worldwide as an effective and reliable tool for early detection of several obstetric adverse outcomes including those described in endometriosis (i.e., pre-eclampsia, fetal growth restriction), the aim of our research was to assess if it could be a useful tool in obstetric care also of endometriosis pregnancies [Lines 67-71]. Based on those evidences, we believed that the identification of a biomarker could be an extremely useful tool in clinical practice to monitor endometriosis pregnancies and perhaps to predict those adverse outcomes. More so, we expanded the sections providing hypotheses to explain our findings including also that, remarkably, endometriosis is characterised by increased levels of TGFB, TNF-α and other cytokines potentially capable of determining defective placentation and placental dysfunction [Lines 297-302].
Question:
Line 75
Please clarify how the authors diagnose severe endometriosis by using ultrasonography. Please show the number of histologically confirmed endometriosis.
Reply: According with ESHRE guidelines issued on February 2022, no unique recommendation on the surgical treatment of endometriosis patients seeking pregnancy are currently available. In this setting, despite the majority of patients in out cohort [57.4%, Lines 195-197] underwent a previous laparoscopic and histologic confirm of the disease, in the remaining cases deep or ovarian endometriosis were assessed before conception by US (yet performed only by skilled operators in this field), We have stated that this was a limitation of our study (see Discussion section). However, a large Cochrane (Nisenblat et al., Ref 59) have extensively demonstrated that TVUS assessment for deep and/or ovarian endometriosis is a reliable diagnostic tool for those lesions. Remarkably, in the Cochrane of Nisenblat et al., a variety of US signs have been validated to describe those lesions. The latter were used by the expert operators in our Institute to detect those lesions by US.
Question:
Did the authors examine the effect of non-severe endometriosis on the value of UtA-PI?
Reply: In our study cohort we included solely patients carrying large ovarian and/or deep endometriosis lesions. For patients treated surgically (57.4%), r-AFS score calculated from surgical reports was at least equal to 16 (i.e., the minimum for defining at least moderate endometriosis), whereas for those with a US diagnosis of endometriosis, either a single ovarian lesion larger than 3 cm, multiple endometriomas, large deep nodules and/or a posterior cul-de-sac obliteration was reported. According with r-AFS staging system, moderate-severe endometriosis is mainly a result of those lesions. Additionally, since US is not able to detect superficial peritoneal endometriosis with enough accuracy that it would be suggested to replace surgery and none of the patients surgically treated in our cohort had solely this endometriosis phenotype, we have repetitively stated in our manuscript that our results may not be applicable for patients carrying those less severe form of endometriosis. Thus, wheatear non-severe endometriosis could also have a measurable impact on placentation reflecting in impaired UtA-PIs should be eventually proven by future research.
Additional comments:
I think the topic of this study is relevant and interesting. However, the authors need to clarify why the authors conducted this study.
Compared with other published material this topic has not been determined and the results of this study are unique.
The methodology appeared to be appropriate.
Yes, the conclusion is consistent with the evidence and argument presented. The authors discussed the main question well.
Reference wise, the authors need to add some recent studies (systematic review) regarding the endometriosis.
The tables and figures appeared to be appropriate and easy to understand.
Reply: As the reviewer can observe reading the reference list, our group performed extensive research on the topics of endometriosis, pregnancy outcome and uterine arteries Doppler studies in pregnancies. We warmly thank the Reviewer to have appreciated our commitment in following a rigorous methodology and our effort in investigating this novel topic.
Round 2
Reviewer 2 Report
The authors have addressed all of my previous comments. Great job.